# Views and Challenges of COVID-19 Vaccination in the Primary Health Care Sector. A Qualitative Study

**DOI:** 10.3390/vaccines11040803

**Published:** 2023-04-05

**Authors:** Maria Moudatsou, Areti Stavropoulou, Michael Rovithis, Sofia Koukouli

**Affiliations:** 1Social Work Department, School of Health Sciences, Hellenic Mediterranean University, GR-71410 Heraklion, Greece; 2Laboratory of Interdisciplinary Approaches for the Enhancement of Quality of Life, Hellenic Mediterranean University, GR-71410 Heraklion, Greece; 3Institute of Agri-Food and Life Sciences, Hellenic Mediterranean University, GR-71410 Heraklion, Greece; 4Nursing Department, School of Health and Care Sciences, University of West Attica, GR-12243 Athens, Greece; 5Faculty of Health, Science, Social Care and Education, Kingston University, Kingston upon Thames KT1 1LQ, Surrey, UK; 6Department of Business Administration & Tourism, School of Economics and Management Sciences, Hellenic Mediterranean University, GR-71410 Heraklion, Greece

**Keywords:** vaccination, COVID-19, adherence, primary care, health professionals, qualitative study, Greece

## Abstract

COVID-19 has affected the global community as it has severely raised population mortality and morbidity rates. Vaccination was seen as a mechanism against the spread of the pandemic. Yet, there are still several reservations about its adoption. Professionals in the field of health care have a crucial frontline role. The present study uses a qualitative research approach to examine Greek health professionals’ views on vaccination acceptance. According to the key findings, health professionals largely accept vaccination. The main reasons cited were scientific knowledge, a sense of obligation to society, and protection from disease. However, there are still numerous restrictions to adhering to it. This is due to the lack of knowledge of certain scientific disciplines or to misinformation, as well as to religious or political convictions. The issue of trust is central to the acceptance of vaccinations. According to our research, the most adequate strategy to enhance immunization and ensure that it is widely accepted is to promote health educational interventions for professionals working in primary care settings.

## 1. Introduction

The COVID-19 pandemic has affected communities across the world at the health, social, and economic levels leading to high mortality and morbidity rates [1]. The use of vaccines was deemed imperative to both reduce the spread of the virus and to limit the number of deaths [2,3]. Vaccination against COVID-19 started in European Union countries on 27 December 2020 and health professionals, as frontline workers, were prioritized [4].

Although vaccination was considered the sole means of protection against the virus, citizens of the European Union appeared to be reluctant and expressed reservations about the use of vaccines. Among the countries with low vaccination rates were France, Greece, Slovenia, and Italy [5,6].

The World Health Organization considers vaccination hesitancy to be one of the ten greatest threats to public health [7]. There are many and complex reasons for the non-acceptance of vaccines. These reasons vary from country to country, but also from region to region within the same state and they can also fluctuate over time [8]. The Health Belief Model (HBM) is one of the theoretical models that attempt to explain the reasons for adopting specific health behaviors, such as health prevention and the use of vaccines [9]. Personality characteristics, the intention to adopt an action (intention) and the ability to complete it (self-efficacy), the likelihood that a health problem will occur to us (perceived susceptibility), the perceived benefit of certain health behaviors (perceived benefit), and subjective norms (subjectivity norms) impact health behaviors [10,11,12]. For instance, the Health Belief Model (HBM) has been instrumental in interpreting the acceptance of vaccination against influenza [12].

Although the vaccine against COVID-19 has been considered essential to deal with the pandemic, there are still several objections to its use [13]. Several reasons for refusing or accepting vaccines have been documented [13], including the trust that one has in a country’s social and economic system, or in the existing health system, health professionals, or science [6,8]. It also appears that mistrust of vaccines has social and political ramifications. It is often an occasion for expressing social reactions as well as professional and union claims [14,15].

In particular, mandatory or non-mandatory vaccination has always been an issue for many vaccine categories. On the one hand, there are strong supporters of vaccination who consider it a measure to protect society and public health, while those who disagree associate it with a violation of personal freedom [16]. Many times, however, the health professionals themselves question the use of vaccines, which is related to a general lack of trust in the health system of their country. In addition, even though not all health professionals have the same level of scientific knowledge about vaccination, this does not prevent them from openly expressing their reservations about the use of vaccines, this way influencing the preventive behavior of the general population [17]. 

The level of existing knowledge and information about vaccines influences their acceptability. When there are gaps in scientific knowledge, conditions are created that encourage the propagation of false information and erroneous beliefs. Additionally, the scientific knowledge related to COVID-19 is still evolving, and, frequently, what is already known is insufficient to close misinformation gaps and eliminate misconceptions, hence reducing preventive actions taken by both health professionals and the general population [16,17,18]. Health professionals have a crucial role in disseminating information, providing health education, and attempting to convince people to follow vaccination schedules [3,19,20]. Because of this, the prevalence of vaccination and its acceptance by the general public are significantly influenced by the views and personal vaccination decisions of health professionals [3,19,21].

Research has shown that health professionals who follow vaccination procedures are more likely to recommend them it to friends, family, and patients, to reduce any resistance and to create a climate of trust in vaccines [6,22,23,24]. It is thus important that health professionals receive the correct information and acquire the relevant scientific knowledge about how vaccines work in order to be able to support them and eliminate all the reservations people may have due to ignorance [17]. As some groups of health professionals, such as physicians, are considered more receptive to vaccination compared to nurses or social workers [25], the provision of tailor-made information [26] and the in-depth exploration of their views on vaccination are considered to be of significant importance.

## 2. Methodology

### 2.1. Aim

The aim of the study was to examine, in-depth, health care professionals’ views, attitudes, and perceptions regarding vaccination against COVID-19.

### 2.2. Design

To understand and explore the views of professionals and the factors contributing to COVID-19 vaccination uptake, the Health Belief Model (HBM) was used [9,11]. The HBM was initially developed to predict health and preventive behaviors. The different constructs of the model include ‘perceived susceptibility’ and ‘perceived seriousness’ (leading to the ‘perceived threat’ from a disease) and the perceived benefits (minus perceived barriers) from taking a preventive action (e.g., vaccination). Additionally, perceived threat from a disease is affected by ‘sociodemographic and psychosocial variables’, ‘cues to action’, including prompts by the health personnel, and self-efficacy, i.e., the level of a person’s confidence in his or her ability to successfully perform a behavior. Based on this conceptual framework, the research team focused on specific factors of the model to explore the challenges of COVID-19 vaccination.

For this purpose, a qualitative research design was used, based on the principles of content analysis. Qualitative research is used when an in-depth analysis of complex psycho-social phenomena is attempted. It is considered the most appropriate research approach for investigating human experiences, feelings, attitudes, and perceptions. Qualitative studies take place within the participants’ natural environment, while the researcher’s active role in the study provides an in-depth understanding of the participants’ perspectives, as well as a comprehensive and thorough interpretation of the phenomenon under investigation [27,28,29].

### 2.3. Study Population and Sample

Health professionals working in primary health and social care services on the island of Crete, Greece, such as Primary Health Centers (PHC), Local Health Units (LHU), and Home Assistance Programs (HAP), comprised the study population. PHC and the LHU (the Greek acronym is TOMY) offer services to the general population of their reference area. PHC manage more complex health conditions and provide specialized prevention services, diagnostic tests, maternal and childcare, dental services, and surgical treatment of incidents. Additionally, PHC direct cases upon assessment for further appraisal at the secondary or tertiary health care level. Some PHC are specialized to address mental health needs at the community level. Local Health Units (called TOMY) support a range of health services, namely preventive health care, scheduling periodic tests, holistic patient care and treatment of chronic diseases, referrals to health care services, handling urgent non-serious issues, and public services such as vaccination and homecare. Finally, HAP focus on elderly people living alone and people with disabilities, delivering health and social care services at home for these groups. All these units are staffed by multi-disciplinary health care teams and are Community-based Health Care Services aiming to provide holistic and high-quality patient-centered inclusive care, dismantling concurrent barriers to health care access. A purposive sampling strategy was used. This strategy results in a sample of people who meet the researcher’s predetermined criteria. It is referred to as a strategic method of sampling individuals related to the research question and enables the researcher to identify those participants who are most likely able to provide a wide range of responses on the subject under investigation [29,30]. This strategy is used, mainly in qualitative research, when the researcher aims to identify participants with specific characteristics and cases that are important for the study topic in terms of gaining wealth and depth of information and data [31]. In our study, the inclusion criteria were being health care professionals from diverse specialties and having more than five (5) years of working experience in primary health care settings. These criteria were considered essential for obtaining adequate information, representative of the health care professionals’ views on COVID-19 vaccination in the primary health care sector. In this respect, the applicability of the study findings in similar settings was supported and enhanced. Furthermore, our sampling strategy attempted to proportionally represent all critical groups of health professionals who work in primary health care settings on the island of Crete. As such, the total number and the distribution of the Primary Health Centers and the Local Health Units within the region of Crete were considered and study sites were selected as representative as possible. Evenly, health professionals from the Home Assistance Programs participated in the study. Additionally, although in primary health care settings guiding principles in the interdisciplinary teams are common, we also tried to have a sample that was fairly evenly split between professionals with a more medical background and knowledge (physicians, nurses, health visitors), addressing the medical aspects of care, and professionals with a more social background and knowledge (social workers, sociologists), managing the social aspects of a person’s care. Therefore, our final sample consisted of 11 persons from the first group (7 physicians, 3 nurses, and 1 health visitor) and 15 persons from the second group (11 social workers, 2 sociologists, and 2 family assistants).

The participants’ recruitment was supported by the Heads of the Departments of the involved health care facilities. After advertising the study, through informal meetings and networks, the potential participants were contacted by the researcher, and they were informed about the purpose and nature of the proposed study. Informed consent was attained before the commencement of the research.

The sample consisted of twenty-six (26) health professionals, from various specialties and different scientific backgrounds, i.e., social workers, nurses, general practitioners, sociologists, health visitors, nurse assistants, and family assistants (Home Assistance Programs employees who assist with household maintenance). The sample’s socio-demographic characteristics are described in Table 1.

### 2.4. Data Collection

Semi-structured interviews were conducted to collect the data. The semi-structured interview is the most appropriate data collection tool for examining individuals’ viewpoints, beliefs, and perceptions of particularly sensitive issues [32]. Through a semi-structured interview, the researcher is able to maintain some control over the content of the dialogue, but, at the same time, the participants have the opportunity to express themselves freely and discuss their views and experiences in depth [33]. The development of the interview questions was based on the relevant literature on COVID-19, the Health Belief Model (HBM), and the available information on the course of vaccinations. Interview questions focused on the following axes: vaccination and health professionals, vaccination compliance, barriers to vaccination, and suggestions for the future.

The interviews were conducted by one member of the authoring team in a methodical manner and at a time chosen by each participant, using a synchronous online video conferencing technique. Despite the fact that face-to-face interviewing is considered the gold standard in qualitative research, online interviewing has recently been increasingly applied [34]. Conducting online interviews became a necessity especially during the era of the pandemic, when social distancing was mandatory [35]. This type of interview was valued for eliminating stress and supporting both the researcher and the participant to express themselves freely and conveniently [34,36].

In the present study, the place and time of the videoconferencing supported the effective conduct of the interview and ensured the confidentiality of the discussion and the data provided. Before the commencement of each interview, the participants were informed about the recording of the data and written permission was granted for the use of the tape recorder. Interviews lasted between 20 and 60 min. At the end of each interview, the data were transcribed word by word and converted into written text. A code was given to each interview which was used throughout the data analysis process to maintain anonymity. Data collection was completed when data saturation was reached [37], meaning that after completing the twenty-fourth interview, the information provided by the participants was repeated and no new data emerged that would necessitate further coding.

### 2.5. Data Analysis

Framework analysis was used to analyze the data derived from each interview. This method is one of the many analytical techniques used in qualitative research and belongs to the broad group of methods of analysis called thematic or qualitative content analysis [37]. When analyzing qualitative health research data, framework analysis is an effective technique that is frequently employed. It is a suitable method for interdisciplinary research groups as it offers clear steps in the analysis of textual data derived mainly from semi-structured interviews [37,38]. The interdisciplinary collaboration that this method entails utilizes the dynamics and reflection of the research team, thus enhancing the credibility and the relevance of the study findings [37]. This approach was determined to be the best possible option for assessing the data of the present research.

The framework analysis process entails five distinct, but interconnected, steps: (a) familiarizing the researcher with the data, (b) extracting codes that describe the information contained in the data, (c) extracting broader categories and individual themes, (d) synthesizing the data based on emerging themes, and (e) writing up the results in a coherent way that makes sense of the data [38]. Three members of our research team assessed the coding, processing, and interpretation of data (analyst triangulation) to ensure the trustworthiness of the study results (see Section 2.7 Trustworthiness of the study) [39]. Throughout the process of data analysis, regular meetings were held among the analysts in order to reach a consensus in the analysis and interpretation decisions. The formulation of final themes and subthemes was reached following the full agreement of the analysts.

### 2.6. Ethical Considerations

Before the commencement of the study, permission was granted by the University’s Bioethics Committee (Ref. Approval no. A.A. 94, A.P. 62/23 June 2021). Following that, permissions were granted by the main involved study sites (Home Assistance Programs Ref. Approval no. A.A. 33854 13 August 2021 and 7th Health Region of Crete, Ref. Approval no. A.P. 422 21 October 2021). The participants received timely information about the purposes of the research from the researcher and an e-mail was sent to them in order to provide their consent to participate. Particular emphasis was placed on their voluntary participation in the study. Participants were asked to speak freely about the subject under investigation without any time pressure. They were also informed that they had the right to withdraw from the study at any time, without any penalty. Issues of data confidentiality and anonymity were ensured throughout the study process and the participants were informed that the interview data would be exclusively used by the researcher for scientific purposes only.

### 2.7. Trustworthiness of the Study

The credibility of the present study was ensured by using two techniques, analyst triangulation and member check. The former technique involved three analysts who performed comparative analyses of individual findings by making coding, analysis, and interpretation decisions. Member checking was the second technique used to enhance the credibility of the study and is considered a productive research practice [40]. The term “member” refers to the various participants who hold multiple roles within the context of the study. Feedback was provided from study participants regarding the themes and sub-themes, the interpretations, and the conclusions derived from data analysis and synthesis. Using these techniques, the data and the findings of the present study were strengthened, as they were examined through multiple points of view [39,40]. COREQ guidelines for reporting qualitative research were used to ensure that the main research domains were appropriately addressed [41].

## 3. Findings

Twenty-six (26) health professionals who were employed in various primary health care services participated in the study. These professionals provided care to the general population, to people with disabilities, and to elderly people (Table 1).

The five main themes derived from the framework analysis were: (a) The usefulness and benefits of vaccination for health professionals; (b) Mandatory or recommended vaccination; (c) Vaccination adherence of health professionals; (d) Vaccination non-adherence of health professionals; (e) Lessons for the future. Twelve sub-themes reflecting mutually exclusive concepts and perceptions were also formulated and presented within the main themes accordingly (Table 2).

### 3.1. The Usefulness and Benefits of Vaccination for Health Professionals

The study participants referred to the benefits of vaccination, focusing on issues of preventing the spread of the virus and protecting the public, the users of health care services, and themselves. In addition, being a role model for the people and supporting the continuation of health care services were also mentioned.

#### 3.1.1. Safeguarding the Patients and Ourselves

The protection of health professionals and their families, as well as the parallel protection of their beneficiaries, who are members of vulnerable groups, dominated the majority of comments addressing the advantages of vaccination.

“We protect ourselves… we safeguard the patients. We have an obligation, to safeguard the beneficiaries. It is crucial… to prevent the spread of the virus to other people.”(R6)

“We interact with plenty of people, so that helps. So, not only should we protect ourselves, but also others with whom we interact at work.”(R11)

“I vaccinated to protect myself, my family, beneficiaries, and everyone else…the people.”(R24)

“My patients were a reason for me to get vaccinated. I want to protect them.”(R10)

#### 3.1.2. Being a Role Model

For the general public, health care professionals serve as role models. They promote vaccination among the general population by being vaccinated.

“Health care workers are role models. Demanding vaccination from the other citizens while remaining unvaccinated as health care professionals is impossible.”(R14)

“We provide services for the people. Let’s set an example. Let’s be an example. To show that vaccination is necessary. We see what happens….”(R2)

#### 3.1.3. Maintaining Provision of Care

Health care professionals demonstrated that vaccinations provide the opportunity to maintain the health services’ purpose, so hospitals were able to remain open and were not compelled to close due to staff-related COVID-19 cases.

“It protects us so that we can continue working and we don’t have lost working hours in case a colleague gets sick. (Otherwise) …it would cost the health system and it would have to bring in some from the private sector, on top of the cost of our hospitalization.”(R19)

“If there is a case among the staff, the unit risks being closed. As a primary health unit, we cover a large part of the population for prescription and follow-up. Services in and out of the unit will stop if the unit shuts down, which will have an impact on hundreds of patients.”(R5)

### 3.2. Mandatory or Recommended Vaccination

Responding to the question of whether vaccination should be a mandatory or a recommended action, the participants reflected positive and negative views. Ιssues of effective communication and people’s opinions regarding the benefits of vaccination were also highlighted.

#### 3.2.1. For: The Only Way to Restrict the Pandemic

Those who supported the ‘mandatory’ vaccination of health professionals stated that vaccines remain the cornerstone of the fight against COVID-19 and the only way to restrict its consequences.

“I am positive about the obligation of getting vaccinated. You can’t dispute science. I can’t figure out e.g., if a cancer patient goes to the hospital that the nurse will be unvaccinated. We are neither in a position to lose personnel, nor to lose lives because we as health workers were not convinced by the vaccine.”(R9)

The study participants believe that the only solution is to require health personnel to get vaccinated, due to the sudden and life-threatening outbreak of the pandemic and the lack of another alternative.

“I believe that the obligatory vaccination is necessary. At this point, we cannot act differently. I have no objection to being compulsory.”(R13)

“I strongly support the obligatory vaccinations not only for health and social care professionals but for teachers, policemen and priest.”(R18)

#### 3.2.2. Against: The Right to Personal Choice

Others though defended the “recommended” vaccination by offering numerous justifications. Vaccination was opposed by some who believed that vaccination should remain a personal choice. They claimed that their behavior was justified by moral and ethical considerations.

“I was never against vaccines or any kind of medical procedure. I’m resisting because I feel that my personal will has been threatened. I was interested in learning how the vaccine worked. I wanted to see the results of vaccination. (However) making something mandatory is unconstitutional. The freedom of will and self-determination of Greek citizens was threatened with compulsion.”(R16)

“Any population group should not be subject to obligations. It opposes democracy.”(R8)

“Many experts were concerned. Given their morals and ideals, they believed it to be immoral.”(R12)

Some were against compulsion, considering it an inappropriate method to persuade people because it causes emotional reactions.

“I resented the compulsion. Some colleagues were suspended. Perhaps another way could be found. I’m not in favor of that. Give them some incentives, like less time on-call or other rewards. They will respond out of stubbornness when it is a form of blackmail.”(R3)

### 3.3. Vaccination Adherence of Health Professionals

Adherence to vaccination was associated with scientific knowledge, education, professional background, and responsibility. Maintaining their professional post, fear of contamination and death, and transmitting the virus to the family and loved ones were reported by the study participants as major reasons to be vaccinated.

#### 3.3.1. Scientific Knowledge and Responsibility

The vaccination of health personnel was prompted by their scientific knowledge, according to some respondents. For the study participants, physicians are more easily convinced because they have confidence in the scientific guidelines and believe in vaccines.

“It helps that health staff have adequate health education and knowledge. The majority of health professionals were persuaded. It was easier to persuade doctors. They are more open-minded in learning and accepting new things. Medical thinking understands what a side effect means. Paramedics that work in the health field delayed. For example, stretcher-bearers or ambulance drivers delayed.“(R23)

“The health workers had the knowledge and made the right decision. They see people perishing and this has led them to believe in vaccination. They show confidence in the instructions of science. They believe in vaccines and science.”(R1)

“Health professionals have a sense of duty. They wish to protect the vulnerable populations they work with.”(R6)

#### 3.3.2. Fear of Contamination and Death

Some health professionals may have chosen to get vaccinated due to personal factors, such as fear of illness and death for themselves and their loved ones.

“The fear of getting contaminated. There is no way for us to escape. We are contacting families with COVID-19 patients. It only takes a split second of carelessness to get infected.”(R26)

“I believe that the fear of death and illness had a lot to do with it. I was also scared of getting sick and having a bad experience, or dying. I was also afraid of being sick and the health system being unable to take care of me. The same applies to other healthcare workers. Fear is the overriding feeling. That’s why many of us got vaccinated.”(R4)

“I was afraid to hug my child. There was reluctance.”(R19)

“I was afraid to be in touch with my parents. I want to protect them from COVID-19.”(R20)

#### 3.3.3. Maintain Professional Status

Another reason for compliance was the government’s measure of suspension and ‘unpaid leave’ and the possibility of losing their job.

“Most people complied as a result of government pressure. To enable them to work primarily and then, if necessary, travel, shop… and then the idea that they won’t get sick. But the main reason was, because they were forced.”(R13)

“There is no doubt that money was a major driving force. Any healthcare professional with obligations did not want to be suspended from work and was immunized as a result. However, vaccination of paramedics would be aided more by good health education.”(R15)

### 3.4. Vaccination Non-Adherence of Health Professionals

Non-adherence to vaccination was associated with issues of confidence, religion, and conspiracy. Participants discussed the reasons why many health professionals refused vaccination, referring mainly to issues such as vaccine safety, side effects, and theories of religion and conspiracy.

#### 3.4.1. Lack of Confidence

Various reasons explain health professionals’ views about non-compliance with vaccination. Many professionals have expressed fears and reservations about the side effects of the vaccine and the process by which it was developed.

“Personality plays a big role there. Fear has taken a toll. Some people even fear a DNA change. They worry that the vaccination will result in gene alterations. There has been always a paranoia about the side effects of vaccines. Now, maybe more.”(R17)

“The information given at the first stage of the virus outbreak and the rapid development of the vaccine… Everybody was concerned, even those of us who were vaccinated. Everything happened very quickly… what side effects the vaccine might have in the near or indirect future. It is an untested and unlicensed vaccine. This fear prevailed in those who were not immunized.”(R4)

Others cited social or political reasons for not getting vaccinated.

“There are people who refuse to receive the vaccine due to social or political reasons. In the countryside, I’ve heard people say they have no faith in the current administration system, not even when it comes to the vaccine…and some health professionals have the same point of view.”(R22)

#### 3.4.2. Religion and Conspiracy

Religious beliefs also prevent some health professionals from being vaccinated, and the anti-vaccination movement has influenced a certain number among them.

“But also, for religious reasons some refused vaccination. I have heard that they accept (the theory about…) the devil’s mark. That the vaccine originates from the devil. That the apocalypse has come. Health workers with medical knowledge, nurses and health visitors have been less affected.”(R5)

Depending on their personal, cultural, and scientific viewpoints, some health professionals have subscribed to the numerous conspiracy theories.

“Some (chose not to get vaccinated) due to conspiracy theories. It seems tragic to me that health professional believes that microchips are in vaccines against COVID-19.”(R25)

“(There are professionals who believe that)…Something is hidden underneath…. That there are profits from that. (Vaccines are)… a way to spy on us. Finally, personal factors, the social background, the scientific specialty come into play. There are many specialties with different education and training.”(R5)

### 3.5. Lessons for the Future

Τhe pandemic outbreak and the process of vaccination was an outstanding experience for the health scientists. Lessons for the future involved effective health education policies and improved access to information.

#### 3.5.1. Health Education

Participants applauded the health education policy followed, as they believed that the explanations and the updates given to health professionals and the public during the period of the vaccination campaign were adequate.

“They explained to us how vaccines are made, and the immunization process. This satisfied me completely. And then we had an online update from the Civil Protection Service. All of us in my service were content with the information provided.”(R3)

“I am very positive about it (the health education given). Information campaigns are organized. They update us from the hospital. There are guidelines from International Organizations. Information is available and easily accessible. I believe that the health education provided is adequate, although not exhaustive.”(R9)

#### 3.5.2. Access to Targeted Information

Other participants, however, felt that the information provided was rather confusing and emphasized the importance of communicating clear, comprehensive, and concise information. The quality of the information was also criticized.

“We were overwhelmed with hundreds of instructions every day, making it very challenging for us to follow. For instance, four emails per day, with instructions that were constantly changing and entire downloads in text were impossible to follow. It was not satisfactory. The instructions must be clear.”(R26)

Almost all respondents emphasized the necessity of having access to health information and health education.

“Health professionals need improved information and health education. Not every health practitioner has the same knowledge. There should be better information about the vaccine in general. Even when I was more hesitant about vaccinations, I got the essential answers from my service. Personally, I was convinced. No need for extravagant things. In simple words, to explain to those who are not in the medical specialty.”(R5)

“Everyone needs health education. But for health and social care professionals is a necessity.”(R7)

Additionally, recommendations were made regarding the information that should be provided and the procedure that should be followed.

“Per service, a meeting had to be held either on an individual or group level, with the focus being on information and where the health personnel could voice their opinions. Being the first to get vaccinated, we were concerned. We had to take various factors into account, but we lacked the necessary knowledge and information.”(R21)

“A central information source should be available to health practitioners, not just documents. To allow for conversation and to locally resolve issues. To inform health professionals on the benefits of the vaccine, go over safety concerns and the protective measures. We did not know the instructions as a unit. To be told once a month or every three months that we can meet online and talk.”(R6)

## 4. Discussion

This qualitative study focused on community health care personnel provides essential insight into their perceptions of COVID-19 vaccines, including acceptability, hesitancy, and factors associated with hesitancy in vaccine uptake. The majority of health professionals who participated in the study had been vaccinated and had positive beliefs toward vaccination. Many health professionals worldwide stated that they intended to follow COVID-19 immunization, including countries such as Greece [4,42,43], France, Colombia [4,42,43], and Spain [44]. In general, health professionals were in favor of vaccination because they have the necessary scientific knowledge to convince both themselves and the general population of their benefits [45]. In our study, physicians were referred to as the most easily convinced group of health professionals because they have strong confidence in science and immunization guidelines. However, health care professionals who work in the primary care sector, doctors, nurses, social workers, and health visitors seemed to share similar attitudes and reported positive views towards vaccination. This finding is in contrast with previous research which indicated that nurses show greater resistance to vaccination than physicians [44,46].

Furthermore, the participants stated that despite the generally positive stances toward the COVID-19 vaccine, there are instances where health personnel are vehemently opposed to vaccination. A possible interpretation is that many of these professionals may lack an understanding of the vaccines’ value and have limited scientific knowledge for advocating the vaccination process to the general public [17].

According to our results, political, cultural, and religious beliefs negatively influenced health professionals’ intention to be vaccinated. The reaction to vaccination has a social and political dimension. It can be used to criticize governmental decisions about health and to oppose compulsory vaccination as conflicting with democratic values [16]. Health professionals have doubts about vaccination and the health policy adopted for immunization and prevention due to mistrust of the political health system of their nation, with which they are in continuous conflict [5]. Additionally, we found that religious beliefs and conspiracy theories about the use of COVID-19 vaccines hindered the intention of health professionals to vaccinate. Similar beliefs were found to affect a large proportion of the general population in various countries, such as Great Britain [4].

Our study participants emphasized the value of vaccination for health professionals. Among the reasons cited were the safety of oneself and one’s family, as well as their professional obligation to the vulnerable groups that often receive services from primary health care facilities. The Health Belief Model (HBM) posits that when people are convinced of the seriousness of a disease pandemic, feel threatened by it, and are convinced that taking preventative measures such as getting vaccinated helps, they will be motivated to act accordingly [47,48,49,50].

Trust in science and technology may explain acquiescence to COVID-19 vaccination [51]. Some health professionals, according to our study, did not get vaccinated because of fear. The main reason was the side effects of the vaccine. They believed that because it was developed so rapidly, it had not been tested adequately, and, as a result, any potential negative effects were unknown [16]. Relevant research evidence confirmed that fear was a reason for avoiding vaccination [4,42,52,53].

Some of the study participants were convinced that there should be mandatory vaccination of health care professionals. Others reacted negatively to the compulsion, although they accepted that with the vaccine, they protected themselves and their patients. They argued that this obligation was a form of violation of each individual’s right to self-determination and personal freedom. In fact, mandatory vaccination does not address the causes of vaccine refusal; rather, it tries to reinforce social norms and values while enhancing its universal acceptance [54]. Research findings have suggested that supporting educational initiatives to persuade people who are opposed to vaccination as an alternative to making it mandatory would be more ethically and politically correct [8].

Greece, among other European countries, required health care workers to get vaccinated as a compulsory condition for continuing to work [55]. Some health care professionals in our study complied with the immunization procedure, not because they were convinced of its benefits, but rather due to pressure from the government’s suspension of work penalties. Therefore, they admitted that they merely adhered to the vaccine requirements for financial reasons. Vaccination policies have shifted dramatically during COVID-19. Vaccination mandates, although debatable regarding their effectiveness, have been adopted by several countries worldwide as a tool of last resort to increase vaccination rates [56].

Almost all study participants made suggestions for the provision of systematic information and training of health professionals about immunizations. They stressed the value of reliable qualitative and quantitative information for helping clarify issues and improving vaccination adherence. It appears that health education has a favorable impact on vaccination intention, as long as it is supported by a country’s national and regional health policies [25].

Information and scientific knowledge about how vaccines work strengthen the conditions for their follow-up [26]. According to other studies, mass media plays a role in influencing the intention to vaccinate [26]. Some studies suggest that positive attitudes towards vaccination can be reinforced not only by providing information and knowledge but also by the more active participation of health professionals in health policymaking [5]. Research in the UK has shown an increase in health professionals’ trust in the health system when they themselves had an active role in it and participated in the decision-making process [56].

### 4.1. Strengths and Limitations of the Study

The present study includes health care professionals with different professional backgrounds and specialties, and this provides a more comprehensive and integrated perspective on the topic under investigation. The qualitative data that emerged from this study also provide in-depth insight into how and why health care professionals dealt with the dilemma of vaccination during the pandemic. These views can contribute to the formulation of effective health policy and to the development of scientific knowledge at an international level. However, the study was conducted in a certain geographical area and the findings should be viewed under this limitation. Even more, our findings are definitely influenced by the time period in which the research was conducted in relation to the progress of scientific understanding regarding COVID-19 immunization.

### 4.2. Conclusions/Policy Implications and Recommendations for Future Research

Primary health care professionals in Greece, as elsewhere, have a key role in the primary health care sector and serve as the local population’s initial point of contact with the health system, as they are concentrated on illness prevention, health promotion, diagnosis, monitoring, and treatment [57]. They also play a crucial role in immunization programs, which are a part of preventative health services. They are frequently the initial source of information for peoples’ vaccinations [58]. People in rural and urban regions, particularly, may readily, affordably, and regularly contact primary health care professionals, who are regarded as trustworthy sources of information on vaccines [59]. As a result, they contribute significantly to lowering all forms of vaccination hesitancy and fostering confidence in the vaccine among their community [60]. Hence, efforts should be made to combat misinformation and mistrust in the sources of information in the health care setting, as well as in the community. Building confidence in health care workers, health institutions, and national health agencies is therefore very important, as that is where the critical information comes from [61].

The COVID-19 vaccine has proven to be an effective way to combat the pandemic. The World Health Organization, as well as every state at national, regional, and local levels, must adopt health policies that will fully convince health professionals of the usefulness of the vaccine. This will be accomplished through education and reliable scientific data. In order for health professionals to feel like a part of the decision-making process, they must be involved in all phases of vaccination scheduling, implementation, and information. Participation in the decision-making process will enhance health care professionals’ positive views of vaccination and convince the general population. Political will and adequate funding are required actions at the international, national, and regional levels for promoting and sustaining health prevention activities during health crisis situations.

## Figures and Tables

**Table 1 vaccines-11-00803-t001:** Participants’ socio-demographic characteristics.

Participants’ Code Numbers	Gender	Age	Profession	Educational Background	Years of Employment	Health Care Service
R1	F	42	Social Worker	University/ΜSc	15	Home Assistance Program
R2	F	49	General Practitioner	University/ΜA	12	Primary Health Center
R3	F	36	Health Visitor	University	6	Local Health Unit
R4	F	41	Social Worker	University	17	Home Assistance Program
R5	F	39	Social Worker	University/ΜSc	6	Local Health Unit
R6	M	30	Social Worker	University/ΜSc	6	Local Health Unit
R7	F	31	Nurse	University/ΜSc	6	Local Health Unit
R8	F	39	Social Worker	University	11	Home Assistance Program
R9	F	32	Nurse	University	7	Local Health Unit
R10	F	42	Family Assistant	University	10	Home Assistance Program
R11	F	45	Family Assistant	Primary School	16	Home Assistance Program
R12	F	50	Social Worker	University	23	Home Assistance Program
R13	F	44	Social Worker	University	17	Home Assistance Program
R14	M	52	General Practitioner	University	20	Primary Health Center
R15	F	42	Social Worker	University/ΜSc	18	Home Assistance Program
R16	F	52	Social Worker	University	21	Home Assistance Program
R17	M	62	General Practitioner	University	25	Primary Health Center
R18	M	51	General Practitioner	University/ΜSc	13	Primary Health Center
R19	F	49	General Practitioner	University/ΜSc	12	Primary Health Center
R20	F	48	Sociologist	University	11	Home Assistance Program
R21	F	54	Social Worker	University	24	Primary Health Center
R22	F	41	Nurse Assistant	Technical Institute	10	Home Assistance Program
R23	F	47	General Practitioner	University	18	Primary Health Center
R24	F	45	Social Worker	University	20	Home Assistance Program
R25	M	44	Sociologist	University/ΜSc	18	Home Assistance Program
R26	F	57	General Practitioner	University	28	Primary Health Center

**Table 2 vaccines-11-00803-t002:** Main themes and subthemes derived from data analysis.

Main Themes	A. The Usefulness and Benefits of Vaccination for Health Professionals	B. Mandatory or Recommended Vaccination	C. Vaccination Adherence of Health Professionals	D. Vaccination Non-Adherence of Health Professionals	E. Lessons for the Future
Sub-themes	Safeguarding the patients and ourselves	‘For’: the only way to restrict the pandemic	Scientific knowledge and responsibility	Lack of confidence	Health education
	Being a role model	‘Against’: the right to personal choice	Fear of contamination and death	Religion and conspiracy	Access to targeted information
	Maintaining provision of care		Maintain professional status		

## Data Availability

Data generated during the present study is not possible to be shared due to issues of subjects’ privacy and confidentiality.

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
