# Peer review of "Views and Challenges of COVID-19 Vaccination in the Primary Health Care Sector. A Qualitative Study"

_vaccines, 2023, doi:10.3390/vaccines11040803_

Round 1

Reviewer 1 Report

I suggest changing the title as: "Views and challenges of COVID-19 Vaccination in the Primary Health Care Sector: A qualitative study"

The Introduction is unnecessary long

The Methodology is clear, well-presented and solid. I suggest adding the conceptual framework of this study to illustrate the factors influencing to challenges of COVID-19 vaccination. So readers can easily follow the results.

The Results section is well organized and interesting. 

The authors need to discuss more about challenges of COVID-19 vaccination, particularly in the primary health care sectors.

Reviewer 2 Report

I enjoyed reading the study,  largely because the authors took the time to offer a good description of the qualitative research process.   Overall, I am recommending publication,, however, I would welcome additional support for the sampling process.   As always in the case of small sample research, there is the concern regarding generalizability of the findings. I’m not asking for a larger number of participants, but I would welcome a good explanation of why this is a good sample, and why the obtained results might be applicable to other situations.   Specific comments follow.

·         Sampling - The sample of participants included 11 social workers, 7 physicians, and a smattering of other healthcare workers.  Only 2 nurses were noted and no nurse practitioners or physician assistants.   It would seem that to establish the validity of the purposive sampling, the authors might be advised to provide a more detailed description of how they came to choose this sort of sample.   That is, what specifically were the purposes of this particular sampling strategy?    I appreciated the table, but I would like to know why this grouping of 26 individuals constitutes a good sample from which to generalize.

·         Sampling – It was noted that sample members provided service to a wide range of conditions (general health, disabilities, mental health, elderly). Is a breakdown of these data possible?  My concern is that if there is a heavy concentration of participations from one of the specialty areas, perhaps the findings might not be so generalizable to other areas.   I’m looking for something pretty simple that assures me that the findings are reasonably generalizable.

·         Data collection – no problem with use of video interviews.   COVID changed how things had to be done.    Apologies and/or explanations really unnecessary.

·         Content Analysis – Authors nicely describe 5 elements of content analysis.  Perhaps there is a 6th element, and that would be to use incoming data to challenge earlier observations.  My understanding of qualitative research is that it is a process of ongoing challenges.  Perhaps the authors have simply included this step within their item “d.”

·         Content Analysis – I appreciated the description and use of the triangulation approach.   Does this method lend itself to some sort of mathematical index, perhaps a percent agreement index?

Reviewer 3 Report

The idea of the work is interesting and has a noticeable social orientation. However, the study itself was performed on 23 volunteers, which is clearly below the most loyal requirements for sample representativeness. The authors constantly appeal to the medical community, but the proportion of physicians in the examined cohort was only 9 people (39.1%). All other are social workers who should not be classified as medical empliyees. Most of the article is based only on quotations from interviews and cannot be assessed from a sociological point of view. The authors tried to formulate the motives for accepting or not accepting vaccination. However, a simple enumeration of motives is clearly not enough, and the authors do not provide any quantitative data that can be approximated to society as a whole. An attempt to classify the degree of adherence to vaccination is given in Table. 2. However, how typical these motives are, and most importantly, what role they play in the choice of vaccination or refusing it, remained outside the scope of the study. All these shortcomings significantly narrow the scientific significance of the work, reducing it to a lyrical novel. In epidemiology, methods for assessing adherence to a particular medical procedure and/or manipulation have long existed and have been successfully used. I believe that the authors should significantly rethink the design of the work and the tools for its implementation so that the data obtained can be approximated to the population as a whole or certain groups of the population. It is not possible to carry out such an approximation based on the results of a survey of 23 people. The work needs a complete revision based on the fundamental tools of sociology and epidemiology

Round 2

Reviewer 3 Report

Dear Authors,   You have done a lot of work to improve the information content of your work.   At the same time, the peer-reviewed work, after editing, became even more sociological and, undoubtedly, can be essential for Journal of Health Organization.   My suggestion is to submit an article for publication in a journal, dedicated to the organization of public health.